# Edge Matters: A Predict-and-Search Framework for MILP based on Sinkhorn-Nomalized Edge Attention Networks and Adaptive Regret-Greedy Search

## Abstract

Predict-and-search is increasingly becoming the predominant framework for solving Mixed-Integer Linear Programming (MILP) problems through the application of ML algorithms. Traditionally, MILP problems are represented as bipartite graphs, wherein nodes and edges encapsulate critical information pertaining to the objectives and constraints. However, existing ML approaches have primarily concentrated on extracting features from nodes while largely ignoring those associated with edges. To bridge this gap, we propose a novel framework named SHARP which leverages a graph neural network SKEGAT that integrates both node and edge features. Furthermore, we design an adaptive Regret-Greedy algorithm to break the barriers of the problem scale and hand-crafted tuning. Experiments across a variety of combinatorial optimization problems show that SHARP surpasses current SOTA algorithms, delivering notable enhancements in both solution accuracy and computational efficiency.

## 1 Introduction

Mixed Integer Linear Programming (MILP) has seen a surge of interest in many practical applications spanning from production planning (Pochet & Wolsey, 2006; Wu et al., 2013), supply chain scheduling (Sawik, 2011), and resource allocation (Liu & Fan, 2018; Watson & Woodruff, 2011). As a discrete extension of linear programming, MILP often faces both integer and continuous variables, thereby leading to the NP-hard complexity. As a well-established topic, the significant successes of traditional algorithms over the past decades, such as branch-and-bound (Land & Doig, 2010; 1960) and cutting plane algorithm (Gomory, 1960), have made it possible to near-optimal solutions for MILP. Besides, these algorithms consist of the core of commercial solvers, such as Gurobi (Gurobi Optimization, 2022). However, these algorithms adhere to an iterative paradigm which heavily lacks parallelism, often making them unsuitable for handling large-scale problems in industrial scenarios.

Recently, there has been significant interest in machine learning (ML) algorithms as a viable alternative to solve MILP in a data-centric fashion. The potential advantage lies in developing faster algorithms in practice by exploiting typical patterns found through the analysis of large training instances. They could automatically discover variable assignment strategies to speed up the similar instance from the same distribution. Specifically, the attempts by several well-studied algorithms to solve MILPs are shifting towards two major categories: the former involves ML-based algorithms learning branch and bound strategies for exact solving, while the latter adopts ML-based methods to learn heuristic rules for approximate solving. We will discuss these methods respectively in Section A.1 and Section A.2 in detail.

Most notably, the pioneering work in approximate solving involves the Neural Diving approach (Nair et al., 2020), which formulates the MILP problem as a bipartite graph and treats the problem solving as the prediction of integer variable assignments. However, previous studies have highlighted challenges associated with insufficient representation for MILP problems and the inaccuracy of a single prediction from the model. To address the issue of inadequate representation, numerous strategies have been employed to compensate for this, such as generating higher-quality initial

solutions (Nair et al., 2020), enhancing representational capacity through additional modules (Nair et al., 2020), and adopting more complicated models like SelectiveNet (Geifman & El-Yaniv, 2019). To tackle the issue of inaccurate model predictions, the post-searching process is applied after the Neural Diving method. For instance, a confidence threshold is used to control which variables to assign. Building on the aforementioned successes, the Predict-and-Search framework (Han et al., 2022) is proposed to consider feasibility by introducing a trust-region-like algorithm that allows the algorithms to change certain fixed variables. However, the bipartite graph representation of the MILP contains rich information about constraint coefficients and objective function coefficients, which can potentially compromise solution quality. Previous works draws much attention on the node representation while ignoring the edge representation. Additionally, algorithms that directly predict variable assignments often struggle to satisfy the constraints, whereas most post-searching algorithms rely on the specific structure of the problem to determine the neighborhood search range, which limits their applicability beyond the initial designed scope and requires hand-crafted parameter tuning.

To tackle above challenges, We propose a nevel framework named SHARP (**S**inkhorn-regularized **E**dge **A**ttention with **A**daptive **R**egret-based **P**rocedure). In terms of insufficient representation challenge, we innovatively integrate EGAT (Gong & Cheng, 2019) with probability distribution learning to provide a finer-grained representation of nodes and edges. Besides, we use the Sinkhorn algorithm (Sinkhorn & Knopp, 1967) for feature normalization, which accelerates computation and stabilizes the training process. In terms of prediction inaccuracy challenge, we propose a confidence threshold-based greedy regret search method. Specifically, we fix variables greedily on the basis of the variable assignment probability in proportion, and then allow for the adaptation of the last variable assignment in a flexible manner. This strategy not only enhances solution feasibility and improves solving accuracy but also captures subtle differences in problem structures, demonstrating robust generalization capabilities.

We conducted comprehensive experiments on Combinatorial Auction (CA) and Item Placement (IP) problems, evaluating SHARP against other methods using three metrics: Primal Gap (PG), Survival Rate (SR), and Primal Integral (PI).

Our contributions can be summarized as follows:

- To the best of our knowledge, we are the first to propose the EGAT with Sinkhorn normalization, which more accurately captures node and edge information, thereby enhancing the expressive power of the model and accelerating training while improving learning stability.

- Our proposed adaptive variable assignment strategy enhances solution feasibility and overcomes the limitations of previous work that are bound by scale.

- The SHARP respectively achieves average 24.88% and 5.86% performance improvements to the modern solvers the Gurobi and SCIP on primal gaps, and exceed the performance of the SOTA ML-based algorithm by 17.19%.

## 2 PRELIMINARIES

### 2.1 MIXED INTEGER LINEAR PROGRAMMING

A mixed integer linear programming (MILP) problem is a type of the combinatorial optimization problem. In these problems, some or all of the variables are constrained to take integer values. Formally, a mixed integer linear programming problem $M$ can be defined as

$$\min \boldsymbol{c}^{\top} \boldsymbol{x} \quad \text{s.t. } \boldsymbol{A}\boldsymbol{x} \leq \boldsymbol{b} \text{ and } \boldsymbol{x} \in \{0,1\}^q \times \mathbb{R}^{n-q}, \tag{1}$$

where $\boldsymbol{x} = (\boldsymbol{x}_1, \ldots, \boldsymbol{x}_n)^{\top}$ represents the $q$ binary variables and the $n-q$ continuous variables to be optimized. The vector $\boldsymbol{c} \in \mathbb{R}^n$ is the objective coefficient vector, and $\boldsymbol{A} \in \mathbb{R}^{m \times n}$ and $\boldsymbol{b} \in \mathbb{R}^m$ specify the $m$ linear constraints. A solution $x$ is called feasible if and only if it satisfies all the constraints. In this paper, we focus on the aforementioned mixed binary form because, according to the theory introduced by (Nair et al., 2020), mixed integer linear programming involving general integers can be reduced to the above form.

## 2.2 BIPARTITE GRAPH REPRESENTATION OF MILP

The bipartite graph representation of mixed integer linear programming was first proposed by Gasse et al. In 2019, Gasse implemented a lossless bipartite graph representation of mixed integer linear programming as an input for neural embedding networks, as shown in Figure 1. The $n$ decision variables in mixed integer linear programming can be represented as the set of right-side variable nodes in the bipartite graph, while the $m$ linear constraints can be represented as the set of left-side constraint nodes. An edge connecting variable node $i$ and constraint node $j$ indicates the presence of the corresponding variable $i$ in constraint $j$, with the edge weight being the coefficient of variable $i$ in constraint $j$.

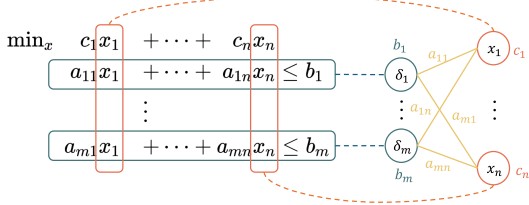

Figure 1: The bipartite graph representation of MILP.

## 2.3 EDGE-ENHANCED GRAPH ATTENTION NETWORK (EGAT)

(Gong & Cheng, 2019) proposed EGAT, which introduces an attention mechanism on graph neighborhoods and fully exploits edge information. This approach is beneficial for learning neural embeddings and model-based initial solution predictions.

Before going into the EGAT, we recall that doubly stochastic normalization. Assuming $\hat{E}$ as the original edge features, the generation process of doubly stochastic normalized features $\hat{E}$ can be described as

$$\tilde{E}_{ij} = \frac{\hat{E}_{ij}}{\sum_{k=1}^{N} \hat{E}_{ik}}, E_{ij} = \sum_{k=1}^{N} \frac{\tilde{E}_{ik}\tilde{E}_{jk}}{\sum_{v=1}^{N} \tilde{E}_{vk}}, \tag{2}$$

where all elements in $\hat{E}$ must be non-negative. It can be easily verified that the normalized edge feature vector $E$ satisfies the following properties:

Furthermore, for an EGAT layer, it can be expressed as follows:

$$\boldsymbol{X}^l = \sigma[\alpha^l(\boldsymbol{X}^{l-1}, \boldsymbol{E}^{l-1})g(\boldsymbol{X}^{l-1})], \tag{3}$$

where $\sigma$ is a non-linear activation function; $\alpha$ is an attention function that returns an $N \times N$ matrix; $W^l$ is a linear transformation; $g$ is a transformation that maps node features from the input space to the output space, typically using linear mapping.

The attention function $\alpha$ can be represented in the form of equation

$$\hat{\alpha}_{ij}^l = \exp\left\{\mathrm{L}\left(\boldsymbol{a}^T\left[\boldsymbol{X}_{i.}^{l-1}\boldsymbol{W}^l\|\boldsymbol{X}_{j.}^{l-1}\boldsymbol{W}^l\right]\right)\right\}\boldsymbol{E}_{ij}^{l-1},$$
$$\alpha^l = \mathrm{DSN}\left(\hat{\alpha}^l\right), \tag{4}$$

where || represents the tensor concatenation operation; DSN is the double stochastic normalization operator described in Equation (2); L represents the LeakyReLU activation function; $W^l$ is the same mapping as in equation Equation (3).

## 2.4 SINKHORN ALGORITHM

The Sinkhorn algorithm is commonly used to compute doubly stochastic matrices, which are non-negative square matrices where both rows and columns sum to 1. (Sinkhorn & Knopp, 1967) stated that a simple iterative method to approach the double stochastic matrix is to alternately rescale all rows and all columns of A to sum to 1. Sinkhorn and Knopp presented this algorithm and analyzed its convergence.

The Sinkhorn algorithm plays a crucial role in various fields such as machine learning, computer vision, and natural language processing, and is frequently applied in graph matching (Wang et al., 2019), image registration (Sander et al., 2022), and optimal transport problems (Groueix et al., 2019). In terms of its functionality for computing doubly stochastic matrices, the Sinkhorn algorithm can be used to accelerate matrix calculations and improve numerical stability.

## 3 METHOD

In this section, we propose a solution framework called SHARP in order to address the problem mentioned in Section 1, which is based on Edge-enhanced Graph Attention Network (EGAT) and mixed-integer programming solver. Subsequently, we conducts a comprehensive computational study comparing this framework with mainstream solvers and existing learning-enhanced methods to demonstrate the superiority of the SHARP framework, which gives an overview in Fig 2.

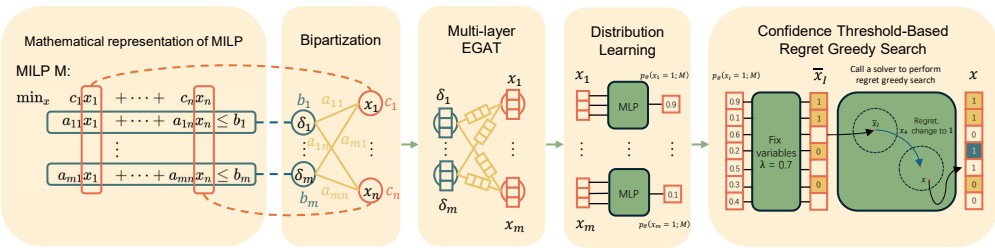

Figure 2: Our framework initially employs a Multi-layer Edge-enhanced Graph Attention Network to encode MILP into embeddings. Subsequently, we utilize Distribution Learning to map each variables into marginal probability distributions. These distributions are then employed in the final stage, where we harness an adaptive Confidence Threshold-Based Regret Greedy Search algorithm to iteratively search for an approximately optimal solution.

### 3.1 SINKHORN NORMALIZATION FOR BIPARTITE GRAPH DOUBLY STOCHASTIC EDGE NORMALIZATION

In EGAT, double random edge normalization is an important component. However, according to Equation (2), it requires E to be an $N \times N$ edge matrix, which poses a challenge for bipartite graphs: for bipartite graphs, the typical edge matrix size is $N \times M$, where $N$ and $M$ are the sizes of the left and right vertex sets, respectively. This makes it impossible to perform calculations according to Equation (2) using the conventional bipartite graph edge matrix. An improved approach is to construct the bipartite graph edge matrix in the form of a general undirected graph, i.e., with an edge matrix size of $(N + M) \times (N + M)$, which can solve the problem of being unable to calculate according to Equation (2). However, this brings another problem: adopting this approach would double the computational cost in terms of both time and space.

To address this problem, we propose introducing the Sinkhorn algorithm, commonly used in fields such as computer vision, to calculate the double random edge normalization for bipartite graphs. Formally, Equation (4) are replaced with the following

$$\hat{\alpha}_{ij}^l = \mathrm{L}(\boldsymbol{a}^T[\boldsymbol{X}_{i\cdot}^{l-1}\boldsymbol{W}^l \| \boldsymbol{X}_{j\cdot}^{l-1}\boldsymbol{W}^l])\boldsymbol{E}_{ij}^{l-1},$$
$$\alpha^l = \mathrm{Sinkhorn}(\alpha^l),$$

(5)

The $\text{Sinkhorn}(x)$ function internally executes the Sinkhorn algorithm, as shown in Algorithm 1. This approach not only significantly reduces the time complexity from $O((N + M)^3)$ to $O(kNM)$ but also limits the space complexity bottleneck to the size of the edge matrix. Notably, following the application of the Sinkhorn algorithm in graph matching, to handle cases where the two sets of nodes in the bipartite graph to be matched have different numbers of nodes, the common practice is to add padding rows to construct a square matrix. After row and column normalization, the padding rows or columns will be discarded.

The specific process of the Sinkhorn algorithm is shown in Algorithm 1.

---

**Algorithm 1** Sinkhorn Algorithm

---

**Input**: Matrix $\boldsymbol{M}$ of size $n_1 \times n_2$
**Parameter**: Iteration count $k$, temperature parameter $\tau$
**Output**: Doubly stochastic matrix $\boldsymbol{S}^{(k)}$

1: $\boldsymbol{S}^{(0)}_{i,j} = \exp(\frac{\boldsymbol{M}_{i,j}}{\tau})$
2: **for** $i = 1$ to $k$ **do**
3:    $\boldsymbol{S}'^{(k)} = \boldsymbol{S}^{(k-1)} \oslash (\mathbf{1}_{n_1}\mathbf{1}_{n_2}^{\top} \cdot \boldsymbol{S}^{(k-1)})$
4:    $\boldsymbol{S}^{(k)} = \boldsymbol{S}'^{(k)} \oslash (\boldsymbol{S}'^{(k)} \cdot \mathbf{1}_{n_1}\mathbf{1}_{n_2}^{\top})$
5: **end for**
6: **return** $\boldsymbol{S}^{(k)}$

---

where $\oslash$ denotes element-wise division, and $\mathbf{1}_n$ represents a column vector of length $n$ with all elements equal to 1.

## 3.2 SKEGAT: EGAT WITH SINKHORN NORMALIZATION

Inspired by Ye et al. (2024), we propose a multi-layer half-convolutional Sinkhorn-normalized edge-enhanced graph attention network. This network combines the advantages of graph convolutional networks with half-convolutions (Gasse et al., 2019) and edge-enhanced graph attention networks (see 2.3), thereby further improving the utilization of edge information. Formally, based on Equation (2) and Equation (5), let $E$ represents the edges in a bipartite graph. The k-layer edge-enhanced graph attention network with a multi-layer half-convolutional structure can be expressed as

$$
\begin{aligned}
\alpha^k_{\boldsymbol{x}_i\boldsymbol{\delta}_j} &= \text{Sinkhorn}(\text{L}(\alpha^T[\mathbf{W}h^{k-1}_{\boldsymbol{x}_i}\|\mathbf{W}h^{k-1}_{\boldsymbol{\delta}_j}])\boldsymbol{E}^{k-1}_{\boldsymbol{x}_i\boldsymbol{\delta}_j}), \\
\boldsymbol{h}^k_{\boldsymbol{x}_i} &= \sigma(\alpha^k(\boldsymbol{h}^{k-1}_{\boldsymbol{x}_i}, \boldsymbol{E}^{k-1})g^k(\boldsymbol{h}^{k-1}_{\boldsymbol{x}_i})), \\
\boldsymbol{h}^k_{\boldsymbol{\delta}_j} &= \sigma(\alpha^k(\boldsymbol{h}^k_{\boldsymbol{\delta}_j}, \boldsymbol{E}^{k-1})g^k(\boldsymbol{h}^{k-1}_{\boldsymbol{\delta}_j})), \\
\boldsymbol{E}^k &= \alpha^k,
\end{aligned}
\tag{6}
$$

where $\boldsymbol{h}^k_{\boldsymbol{\delta}}$ and $\boldsymbol{h}^k_{\boldsymbol{x}}$ represent the hidden state vectors of constraint nodes and variable nodes in the $k$-th layer, respectively; $\alpha^k_{\boldsymbol{x}_i\boldsymbol{\delta}_j}$ denotes the attention coefficient of the edge connecting variable node $\boldsymbol{x}_i$ and constraint node $\boldsymbol{\delta}_j$ in the $k$-th layer; $\boldsymbol{E}^k$ represents the $k$-th layer; $\sigma$ is a non-linear activation function; $g^k$ is a transformation that maps node features from the input space to the output space, typically implemented as a linear transformation.

## 3.3 DISTRIBUTION LEARNING

We use a supervised learning to predict the conditional distribution for MILP instances, following the approach of previous work (Han et al., 2022; Nair et al., 2020). Given a set of MILP instances $\mathcal{M}$ for training, we define the dataset $\mathcal{D} = \{(M, \mathcal{S}^M) \mid M \in \mathcal{M}\}$, where $\mathcal{S}^M$ is the set of solutions for instance $M$. Formally, let $\hat{\boldsymbol{p}}_i \equiv p_{\boldsymbol{\theta}}(\boldsymbol{x}_i = 1; M)$ denote the predicted probability that the $i$-th variable is 1 in instance $M$, parameterized by $\boldsymbol{\theta}$. We define the loss function $\mathcal{L}(\boldsymbol{\theta})$ as:

$$
\mathcal{L}(\boldsymbol{\theta}) = \sum_{M \in \mathcal{M}} \sum_{i=1}^{n_M} \boldsymbol{y}^M_i \log \hat{\boldsymbol{p}}^M_i + (1 - \boldsymbol{y}^M_i)\log(1 - \hat{\boldsymbol{p}}^M_i),
$$

where $n_M$ is the number of variables in instance $M$. The target probability $\boldsymbol{y}_i^M$ is defined as:

$$\boldsymbol{y}_i^M \equiv \sum_{j \in \mathcal{I}_i^M} \frac{\exp\left(-\boldsymbol{c}^{M\top} \boldsymbol{x}^{M,j}\right)}{\sum_{\tilde{\boldsymbol{x}} \in \mathcal{S}^M} \exp\left(-\boldsymbol{c}^{M\top} \tilde{\boldsymbol{x}}\right)},$$

where $\boldsymbol{c}_M$ denotes the objective coefficient vector for instance $M$, $\boldsymbol{x}^{M,j}$ denotes the $j$-th solution in $\mathcal{S}^M$, and $\mathcal{I}_i^M \subseteq \{1, 2, ..., |\mathcal{S}^M|\}$ denotes the set of indices in $\mathcal{S}^M$ where the $i$-th component is 1. This formulation ensures that $\boldsymbol{y}_i^M$ is normalized by the total number of solutions in $\mathcal{S}^M$.

## 3.4 Confidence Threshold-Based Regret Greedy Search

We introduce a novel confidence threshold-based regret greedy search method that leverages marginal probabilities as input. The core concept of this approach is to strategically fix the values of certain variables based on the marginal probabilities generated by a neural network. This technique effectively reduces the problem sizes, consequently mitigating the exponential time complexity bottleneck inherent in the solver. The comprehensive framework of the proposed method is illustrated in Algorithm 2.

The theoretical foundation of our approach lies on the premise that in an optimal solution, variables with a value of 1 should, after neural network processing, exhibit marginal probabilities closer to 1, and conversely, closer to 0. Leveraging this insight to reduce problem size and accelerate the solving process, we employ a strategy to force certain binary variables to take specific values.

Assuming we have determined which variables are to be fixed, we formally define the set of variables to be fixed to 0 as $\mathcal{X}_0$, while the set of variables to be fixed to 1 is represented as $\mathcal{X}_1$. We define a function $\phi(x)$ as follows:

$$\phi(x) \equiv \begin{cases} 0, & \text{if } x \in \mathcal{X}_0, \\ 1, & \text{if } x \in \mathcal{X}_1. \end{cases} \tag{7}$$

A naive approach would be to directly subject these variables to the constraint $\sum_{x \in \mathcal{X}_0 \cup \mathcal{X}_1} |\phi(x) - x| = 0$, resulting in the following problem formulation:

$$\min \ \boldsymbol{c}^\top \boldsymbol{x},$$
$$\text{s.t.} \quad \boldsymbol{A}\boldsymbol{x} \leq \boldsymbol{b} \text{ and } \boldsymbol{x} \in \{0, 1\}^q \times \mathbb{R}^{n-q},$$
$$\sum_{x \in \mathcal{X}_0 \cup \mathcal{X}_1} |\phi(x) - x| = 0.$$

However, this approach still has limitations. If there exists a variable $x \in \mathcal{X}_0 \cup \mathcal{X}_1$ such that $\phi(x) \neq x*$, i.e., there are variable fixation errors, it will lead to deterioration of the solution or even infeasibility. Han et al. (2022) observed that there exists a relatively small $\Delta > 0$ such that $\sum_{x \in \mathcal{X}_0 \cup \mathcal{X}_1} |\phi(x) - x*| \leq \Delta$. To utilize this property, Han et al. (2022) proposed a trust-region-based search method. However, for different problem sizes and structures, this trust-region-based search method lacks generalizability and requires more extensive manual parameter tuning.

Inspired by Yoon (2022), we propose a more generalizable and robust confidence threshold-based regret greedy search method. In the regret greedy idea, undoing previous operations is also considered within the scope of the greedy strategy. In this method, we initially fix certain variables using a greedy approach, while maintaining the flexibility to modify a subset of these fixed variables in subsequent iterations. Introducing a regret mechanism to fix decision errors makes the search process "softer". Formally, A regret coefficient $0 < \lambda < 1$ is introduced, allowing $\lambda|\mathcal{X}_0 \cup \mathcal{X}_1|$ variables to have fixed values different from their corresponding values in the optimal solution during the variable fixation process.

Formally, we define the set of variable which should be fixed to 0 as $\mathcal{X}_0 = \{\boldsymbol{x}_i \mid p_\theta(\boldsymbol{x}_i = 1; M) < 1 - \beta, \ 1 \leq i \leq q\}$, and the the set of variable which should be fixed to 1 as $\mathcal{X}_1 = \{\boldsymbol{x}_i \mid p_\theta(\boldsymbol{x}_i = $

$1; M) > \beta,\ i = 1, 2, \dots, q\}$. The original MILP problem should be subject to the following additional constraint:

$$\sum_{x \in \mathcal{X}_0 \cup \mathcal{X}_1} |\phi(x) - x| \leq \lambda |\mathcal{X}_0 \cup \mathcal{X}_1|. \tag{8}$$

Consider $\mathcal{X}_0$ and $\mathcal{X}_1$ separately:

$$\sum_{x \in \mathcal{X}_0} x + \sum_{x \in \mathcal{X}_1} (1 - x) \leq \lambda |\mathcal{X}_0 \cup \mathcal{X}_1| \tag{9}$$

Consequently, the original MILP problem can be transformed as follows:

$$\min\ \boldsymbol{c}^\top \boldsymbol{x},$$
$$\text{s.t.}\quad \boldsymbol{Ax} \leq \boldsymbol{b} \text{ and } \boldsymbol{x} \in \{0,1\}^q \times \mathbb{R}^{n-q},$$
$$\sum_{x \in \mathcal{X}_0} x + \sum_{x \in \mathcal{X}_1} (1 - x) \leq \lambda |\mathcal{X}_0 \cup \mathcal{X}_1|.$$

As a result, we proceed to solve the new MILP problem described above.

---

**Algorithm 2** Confidence Threshold-Based Regret Greedy Search Algorithm

---

**Input**: MILP instance $M$, neural network probability distribution prediction result $p_\theta (x_i = 1; M)$.
**Parameter**: Confidence threshold $\beta$, regret coefficient $\lambda$.
**Output**: Solution $\boldsymbol{x}$ of the MILP instance $M$.

1: $M' \leftarrow M$
2: $\mathcal{X}_0 \leftarrow \varnothing$
3: $\mathcal{X}_1 \leftarrow \varnothing$
4: **for** $i = 1$ to $q$ **do**
5:    **if** $p_\theta (x_i = 1; M) < 1 - \beta$ **then**
6:       $\mathcal{X}_0 \leftarrow \mathcal{X}_0 \cup \boldsymbol{x}_i$
7:    **else if** $p_\theta (x_i = 1; M) > \beta$ **then**
8:       $\mathcal{X}_1 \leftarrow \mathcal{X}_1 \cup \boldsymbol{x}_i$
9:    **end if**
10: **end for**
11: create constraint $\sum_{x \in \mathcal{X}_0} x + \sum_{x \in \mathcal{X}_1} (1 - x) \leq \lambda |\mathcal{X}_0 \cup \mathcal{X}_1|$ to $M'$
12: $x = \text{solve}(M')$
13: **return** $x$

---

## 4 EXPERIMENTS

### 4.1 EXPERIMENTAL SETUP

#### 4.1.1 DATASET

We evaluate our approach on three MILP benchmark problems: Set Covering (SC), Minimum Vertex Cover (MVC), and one non-graph problem, Combinatorial Auction (CA). For SC, we generate instances with . For CA, we generate instances with 300 items and 1500 bids . These problem types were selected as our benchmarks due to their widespread recognition (Gasse et al., 2019; Han et al., 2022; Ye et al., 2024) and the significant challenges they pose to current off-the-shelf solvers. Besides, We do not use Maximum Independent Set (MIS) and Workload Balance as our benchmark problem following Han et al. (2022) since these two problem is too easy for our proposed approach. The details of the four problems will be provided in Appendix C.

### 4.1.2 BASELINE

In this paper, we conduct comprehensive computational experiments to evaluate the effectiveness and efficiency of our proposed SHARP framework. Our experiments compare our method against a range of established baselines, including 1. traditional MILP solvers such as SCIP and Gurobi, and 2. a state-of-the-art machine learning-based approach, specifically the Predict-and-Search framework proposed by Han et al. (2022).

### 4.1.3 EVALUATION METRIC

**Primal Gap** The Primal Gap $\gamma(t)$ at time $t$ is the relative difference between the optimization objective value achieved by the algorithm being evaluated at time t and the pre-computed known best objective value $f(x*)$ (Berthold, 2006). Formally, it can be expressed as

$$
\gamma(t) = \begin{cases} 1, & \text{if no feasible solution} \\ & \quad \text{at time } t, \\ \frac{|f(x_t) - f(x^*)|}{max\{|f(x_t)|, |f(x^*)|\}}, & \text{otherwise.} \end{cases}
$$

**Survival Rate** The Survival Rate $S(t)$ at time $t$ refers to the proportion of instances where the Primal Gap is below a specified Primal Gap threshold at time $t$, among all instances (Sonnerat et al., 2021).

**Primal Integral** The Primal Integral at time $T$ refers to the integral of the Primal Gap over the interval $[0, T]$ at time $T$ (Achterberg et al., 2012). It reflects both the quality and speed of finding a solution. This metric measures the area under the Primal Gap curve during the solver's solution process, which is equivalent to the integral over time of the Primal Gap of the best feasible solution found so far. Formally, it can be described as

$$
P(T) = \int_{t=0}^{T} p(t)dt = \sum_{i=1}^{I} \gamma(t_{i-1}) \cdot (t_i - t_{i-1}).
$$

### 4.2 RESULTS

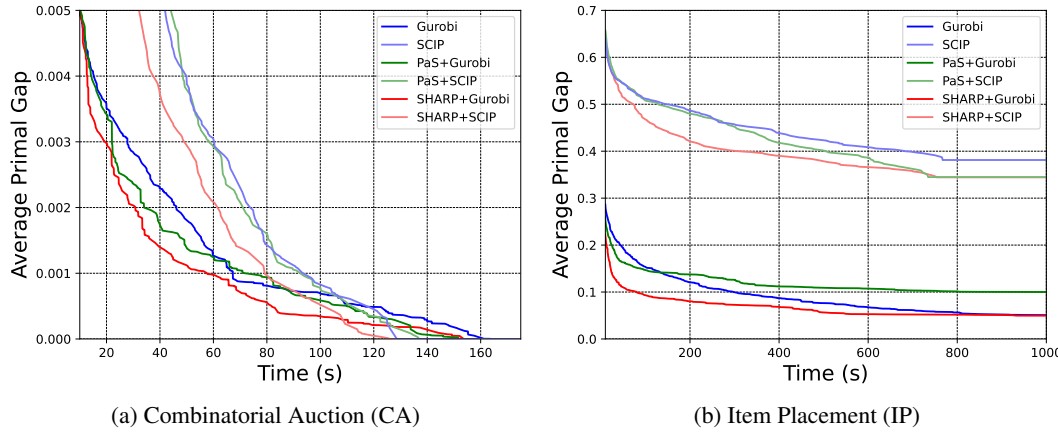

(a) Combinatorial Auction (CA)  (b) Item Placement (IP)

Figure 3: Performance comparison across SCIP, Gurobi, PaS and SHARP, where the y-axis indicates the average relative primal gap based on 50 instances about CA and IP.

Since our method is a Predict and Search strategy similarity to Han et al. (2022), we also utilized Gurobi and SCIP as the benchmark solvers. Figure 3 exhibits the progress of average gap on 50 instances as the solving process proceeds, and Figure 4 show the progress of survival rate with specific primal gap threshold on 50 instances when solving MILPs. In Figure 3a and Figure 4a, we observe that our proposed SHARP outperforms PaS and off-shelf solvers, regardless of whether

Gurobi or SCIP is used as solver. In Figure 3b and Figure 4b, our , even though our proposed SHARP surpass PaS and off-shell solver, combined with whether Gurobi or SCIP.

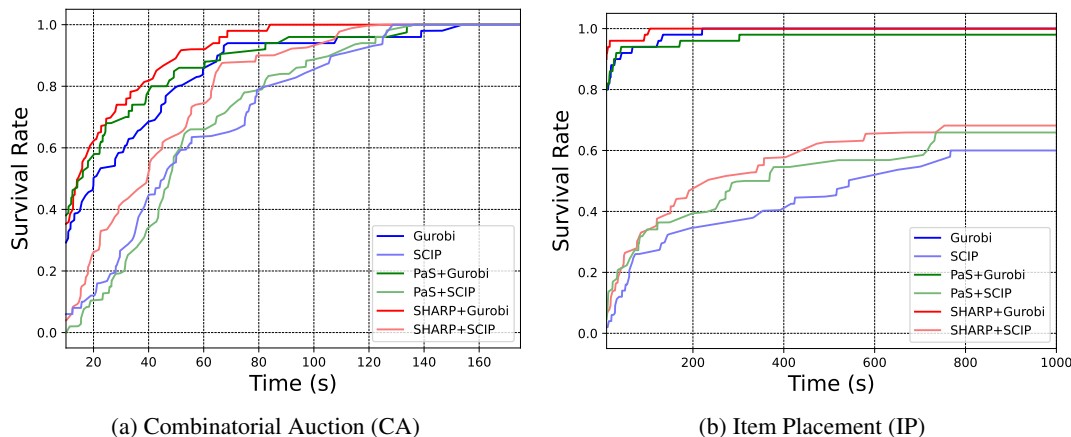

(a) Combinatorial Auction (CA)                      (b) Item Placement (IP)

Figure 4: Survival Rate of performance comparison across PaS, Sharp, Gurobi and SCIP based on 50 instances about CA and IP.

Besides, we also evaluate SHARP on the primal integral on all datasets across different combinatorial optimization problems, which refer to Table 1 in detail. The primal integral in Table 1 is more smaller more better. Similar to the primal gap to the optimal solution, it also shows the trend that SHARP achieves powerful results to the SOTA ML-based algorithms and the modern MILP solver like Gurobi and SCIP.

Table 1: Average Primal Integral on each dataset.

| Solver | Method | CA | IP |
|--------|--------|--------|----------|
|        | SCIP   | 1.0351 | 438.4543 |
| SCIP   | PaS    | 1.0835 | 418.1390 |
|        | SHARP  | **1.0173** | **394.5655** |
|        | Gurobi | 0.2934 | 93.1101  |
| Gurobi | PaS    | 0.2765 | 120.5288 |
|        | SHARP  | **0.2355** | **69.6635** |

## 4.3 ABLATION STUDY

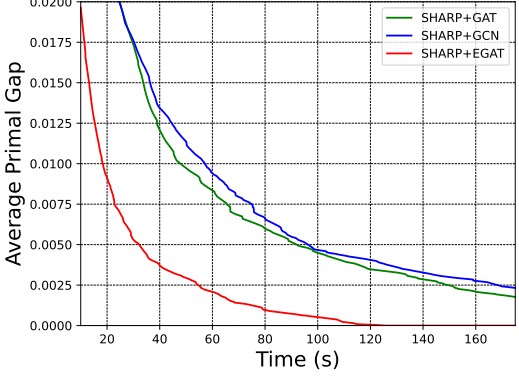

Figure 5: Average primal gap slot

To further instigate the superiority of our proposed SKEGAT in MILP solving, we also conduct experiments against other common GNN. The results are presented in Figure 5, where SHARP+GAT, SHARP+GCN, and SHARP+SKEGAT represent the performance of our proposed method using GAT, GCN, and SKEGAT as neural network, respectively.

## 5 CONCLUSION

This paper introduces SHARP, a novel framework that addresses the limitations of existing approaches by integrating EGAT with Sinkhorn normalization to enhance the representation of both nodes and edges within MILP problems. This combination not only provides a richer and more accurate representation of the problem space but also facilitates faster convergence and improved stability during the training phase. Furthermore, the proposed adaptive variable assignment strategy, which incorporates a confidence threshold-based greedy regret search method, significantly enhances the feasibility of the solutions generated by the model. However, the SHARP still has some shortcomings, such as further integrating with the pos-search search algorithm to inversely optimize the parameters. Nevertheless, the SHARP represents a significant step forward in the application of machine learning techniques to MILP problems.

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

# A  RELATED WORK

In general, ML-based algorithms typically follow two primary strategies: those that rely on branch and bound algorithms for exact solutions, and those that use heuristic algorithms for approximate solutions. Zhang et al. (2023) provides a comprehensive survey of solving MILP via machine learning.

## A.1  BRANCH AND BOUND

As a famous topic in operation research, the branch and bound (BnB) algorithm (Land & Doig, 2010; 1960) aims to iteratively partition the solution space through branching and pruning until an exact solution is built. These algorithms form the foundation of most modern MIP solvers (Gurobi Optimization, 2022; Bolusani et al., 2024), typically alongside with cutting plane algorithms (Gomory, 1960).

Most ML-based methods focus on locally improving the selection strategies within the BnB algorithm, such as branching variable selection, node selection, and cutting plane selection. These methods frame the selection strategies as decision-making problems (Khalil et al., 2016; Balcan et al., 2018) and develop heuristic selection rules by learning from extensive sets of training instances.

Branching variable selection has been drawn much attention due to its significant impact on BnB performance. There has been developed various branching rules for variable selection, including strong branching (SB) (Applegate et al., 1995), pseudo-cost (Bénichou et al., 1971), and hybrid branching (Achterberg & Berthold, 2009). Recent approaches have employed imitation learning (Ho & Ermon, 2016) to approximate these branching rules. Marcos Alvarez et al. (2014) and Khalil et al. (2016) learned branching policies by imitating the strong branching rule. To mitigate the complexity of feature calculation, Gasse et al. (2019) proposed encoding the MILP into a lossless bipartite graph, which serves as input to a graph convolutional network (GCN) (Kipf & Welling, 2016). Building upon this bipartite graph model, Nair et al. (2020) employed a GCN to encode MIPs as bipartite graphs and train it to imitate an ADMM-based policy for branching. Gupta et al. (2020) introduced a hybrid architecture that combines GNN and MLP models. Deviating from the bipartite representation, Ding et al. (2020) generated a tripartite graph from MIP formulation and train a GCN for variable solution prediction. Additionally, Zarpellon et al. (2021) proposed a novel neural network design that incorporates the global BnB tree state.

Compared to the branching variable selection, other strategies receive less attention. He et al. (2014) employed imitation learning to train both node selection and node pruning policies, while Yilmaz & Yorke-Smith (2021) focused more on learning a policy to select only the most promising children of a given node. And for cutting plane selection, Tang et al. (2020) introduced an MDP formulation and trains a reinforcement learning agent using evolutionary strategies. Huang et al. (2022) proposed a cut ranking method using neural networks in a multiple instance learning setting.

## A.2  ML-BASED APPROXIMATE MILP SOLVING

While BnB can solve MILP instances exactly, its NP-hard nature makes it impractical for large-scale problems. Therefor, researchers have explored heuristic-based methods to find approximate solutions efficiently.

One promising direction in this field is the application of machine learning to generate initial solutions or partial assignments for MILP problems. Nair et al. (2020) introduced Neural Diving, a learning-based approach that uses GCN to generate promising partial assignments for integer variables. They then trained an additional network called SelectiveNet (Geifman & El-Yaniv, 2019) to determine which variable assignments to keep. Yoon (2022) simplified this framework by introducing a confidence threshold concept. Building on these efforts, Han et al. (2022) proposed a Predict-and-Search framework, incorporating a trust region-like algorithm to enhance feasibility by allowing the solver to modify certain fixed variables. In our paper, we primarily follow this line of research due to its promising potential.

Inspired by the success of Neural Diving, researchers have applied similar machine learning techniques to enhance traditional heuristics for MILPs, particularly Large Neighborhood Search (LNS) and Feasibility Pump (FP). LNS iteratively explores neighborhoods of the current solution, while

FP alternates between finding rounded solutions to the continuous relaxation and nearby feasible integer solutions. The effectiveness of these methods depends crucially on aspects like initial solution generation and search strategy. For LNS, Song et al. (2020) leveraged the ideas from Neural Diving and used imitation and reinforcement learning for neighborhood selection. Similarly, Sonnerat et al. (2021) adapted the neural diving concept for both initial solution and neighborhood selection in LNS. In the FP domain, Qi et al. (2021) employed reinforcement learning to guide non-integer solution selection during the algorithm's execution, further demonstrating the potential of machine learning in improving MILP heuristics.

Recent studies have increasingly concentrated on developing predictive models to intelligently select and utilize existing MILP solvers or methodologies for optimal performance. For instance, Ding et al. (2020) developed a prediction-based method to choose between heuristic algorithms and exact branching approaches for specific variables. Similarly, Grover et al. (2018) proposed an online learning framework using reinforcement learning to select predefined heuristics in CPLEX based on instance features. Taking a data-driven approach, Xavier et al. (2021) trained a k-Nearest Neighbors (KNN) model to predict redundant constraints, good initial feasible solutions, and promising affine subspaces for optimal solutions. Moreover, the concept of early stopping, widely employed in hyperparameter optimization (Makarova et al., 2022), exhibits substantial promise for MILP solvers by predicting the optimal termination point of the BnB process.

## B  HYPERPARAMETER SETTINGS

The hyperparameter settings for each baseline method in this paper are detailed in Table 2.

Table 2: Hyperparameter settings for PaS and SHARP

| dataset | PaS | | | SHARP | |
|---|---|---|---|---|---|
| | $k_0$ | $k_1$ | $\Delta$ | $\beta$ | $\lambda$ |
| CA | 400 | 0 | 10 | 0.95 | 0.05 |
| SC | 1500 | 0 | 100 | 0.96 | 0.02 |
| IP | 400 | 5 | 1 | 0.92 | 0.02 |

## C  DATASET DETAILS

The statistical data of the datasets for each problem are presented in Table 3, including the number of constraints, the number of integer variables, and the number of continuous variables.

Table 3: Maximum problem sizes of each dataset

| dataset | # constr. | # binary var. | # continuous var. |
|---|---|---|---|
| CA | 6,396 | 1,500 | 0 |
| SC | 5000 | 4000 | 0 |
| IP | 195 | 1,083 | 33 |

