# OpenReview forum: "Edge Matters: A Predict-and-Search Framework for MILP based on Sinkhorn-Nomalized Edge Attention Networks and Adaptive Regret-Greedy Search"
_ICLR.cc/2025/Conference — ICLR 2025 Conference Withdrawn Submission_

### Official Review · Reviewer_wGQP · 2024-10-16

**Soundness:** 2
**Presentation:** 2
**Contribution:** 3
**Rating:** 5
**Confidence:** 4

**Summary:**

The paper introduces a novel framework named SHARP for solving Mixed-Integer Linear Programming (MILP) problems, leveraging a graph neural network called SKEGAT that integrates both node and edge features. The key contributions include:

1. The use of an Edge-enhanced Graph Attention Network (EGAT) with Sinkhorn normalization, improving the representation of nodes and edges, which enhances the expressive power of the model while stabilizing training.
2. An adaptive variable assignment strategy that employs a confidence threshold-based greedy regret search method to enhance solution feasibility and scalability.
3. The proposed SHARP outperforms modern solvers like Gurobi and SCIP in terms of solution accuracy and computational efficiency, achieving a significant improvement in primal gaps.

The framework demonstrates superior performance compared to state-of-the-art methods across a range of combinatorial optimization problems.

**Strengths:**

### Originality
The paper's originality is notable in its approach to incorporating both node and edge features into the graph representation of MILP problems using the SKEGAT model, which utilizes Sinkhorn-normalized edge attention. This is a significant deviation from previous methods that have largely ignored edge features, focusing solely on node features. Additionally, the adaptive variable assignment strategy, based on a confidence threshold and regret-greedy search, offers a novel method for improving both the solution feasibility and computational scalability of MILP solutions.

### Quality
The quality of the paper is reflected in the careful development of the SHARP framework, which is grounded in well-established machine learning and optimization techniques. The integration of Sinkhorn normalization to stabilize and improve the efficiency of the learning process demonstrates a strong theoretical foundation. The experimental evaluation includes multiple well-chosen benchmark datasets and metrics, such as primal gap, survival rate, and primal integral, showcasing the comprehensive testing of SHARP against strong baselines like Gurobi, SCIP, and state-of-the-art machine learning-based approaches.

### Clarity
The paper is clearly written, with a logical flow from problem formulation to solution methodology and experimental results. The inclusion of detailed descriptions of existing methods helps contextualize the contributions of the proposed SHARP framework. Moreover, the paper explains the technical details behind the SKEGAT model and the adaptive variable assignment strategy well, making it accessible to readers familiar with graph neural networks and optimization. However, certain parts, particularly those related to the technical implementation of the Sinkhorn normalization and confidence threshold-based regret search, might benefit from additional visual aids or examples to improve clarity further.

### Significance
The significance of the paper lies in addressing a core limitation in the machine learning-based MILP community, specifically the lack of effective edge feature utilization in graph representations. By using EGAT with Sinkhorn normalization, the paper demonstrates significant improvements over existing state-of-the-art methods in terms of solution quality and computational efficiency, especially on large-scale MILP problems. The framework's applicability to practical combinatorial optimization problems such as combinatorial auction and item placement also underscores its broader impact and potential utility in industrial applications. Additionally, the adaptive regret-greedy approach is an important contribution, providing a scalable solution that could be integrated into existing solvers to enhance their performance.

Overall, the paper offers a solid combination of originality, quality, and practical significance, with a clear and well-structured presentation that effectively communicates the innovative aspects of the work.

**Weaknesses:**

### 1. Insufficient Comparison with Diverse GNN Models
**Weakness**: Although the paper introduces the SKEGAT model and demonstrates its effectiveness, the comparison is largely limited to GAT and GCN in the ablation study. This narrow scope misses the opportunity to benchmark SHARP against other cutting-edge graph neural network models, such as Graph Isomorphism Networks (GIN) or Transformer-based graph architectures, which might provide additional insights into the specific benefits of the proposed method.

**Suggestion**: Include more comprehensive comparisons with a broader range of GNN architectures, such as GIN or Graph Transformer models, which have been effective in graph representation tasks. This would further validate the superiority of SKEGAT and provide deeper insights into its strengths and limitations.

### 2. Limited Generalizability to Non-bipartite Graph Structures
**Weakness**: The current approach primarily focuses on bipartite graph representation and normalization techniques for MILP. This focus may restrict its applicability when dealing with non-bipartite or more complex graph structures that are common in real-world MILP problems. The methodology lacks a discussion of how the framework could be adapted or extended to handle these different types of graph structures effectively.

**Suggestion**: Consider including a section on potential extensions of the SHARP framework to handle more diverse graph structures. Discussing how the framework might generalize beyond bipartite graphs and the challenges associated with such extensions would strengthen the paper's scope and practical relevance.

### 3. Sparse Evaluation Metrics
**Weakness**: The evaluation metrics used, such as primal gap, survival rate, and primal integral, provide useful insights but might not fully capture the practical usability and efficiency of the SHARP framework in different settings. For example, the time complexity analysis is provided only in terms of O-notation, without concrete runtime measurements or memory consumption data.

**Suggestion**: Add empirical analysis of runtime and memory usage to better demonstrate the efficiency and scalability of SHARP, particularly for large-scale MILP problems. These practical metrics would make the results more compelling, particularly for industrial applications where resource constraints are a critical consideration.

### 4. Limited Analysis of Hyperparameter Sensitivity
**Weakness**: The paper includes hyperparameters like the confidence threshold (β) and the regret coefficient (λ), but there is no detailed analysis of their impact on performance. Hyperparameter tuning is crucial for machine learning-based approaches, and an insufficient discussion of this aspect could make it challenging for practitioners to apply the framework effectively.

**Suggestion**: Conduct a sensitivity analysis of key hyperparameters (e.g., β, λ) to understand their influence on solution quality and efficiency. Providing guidelines on selecting appropriate values based on problem characteristics would improve the usability and robustness of the framework.

### 5. Lack of Robustness Analysis
**Weakness**: The robustness of the SHARP framework is not thoroughly evaluated, especially under varying problem types or noisy input data. Given the complexity of MILP problems and the use of learned models, SHARP's stability and reliability under different conditions should be assessed.

**Suggestion**: Include an evaluation of SHARP under different problem conditions, such as varying problem sizes, constraint tightness, or input noise. Adding experiments that illustrate how the performance of SHARP changes with different levels of problem difficulty or data quality would strengthen the paper's claims about its robustness and adaptability.

### 6. Missing Intuitive Explanation for Key Concepts
**Weakness**: The explanations for some of the key concepts, such as Sinkhorn normalization and the regret-greedy approach, are technically dense and may be challenging for readers who are less familiar with advanced optimization or graph-based ML methods.

**Suggestion**: Simplify or add intuitive examples to explain concepts like Sinkhorn normalization and the confidence threshold-based regret greedy search. This would improve accessibility, making the paper more approachable for a wider audience, including practitioners who may not be experts in graph-based learning methods.

### 7. Scalability Limitations in Industrial Scenarios
**Weakness**: The scalability of SHARP, while mentioned, is not extensively tested against large-scale industrial datasets beyond the benchmark problems considered. Moreover, the impact of increased computational costs due to EGAT's attention mechanisms on industrial-scale problems is not clearly addressed.

**Suggestion**: To make the contribution more convincing for real-world industrial applications, add experiments on larger, more complex datasets that resemble real-world MILP problems. Explicitly addressing the trade-offs between scalability and computational cost for attention mechanisms would clarify SHARP’s suitability for large-scale industrial use.

**Questions:**

See Weaknesses.

---

### Official Review · Reviewer_gQzU · 2024-10-24

**Soundness:** 2
**Presentation:** 3
**Contribution:** 2
**Rating:** 5
**Confidence:** 2

**Summary:**

This paper proposes SHARP, a novel framework designed to solve Mixed-Integer Linear Programming (MILP) problems by leveraging machine learning techniques, particularly in a Predict-and-Search strategy. Traditionally, MILP problems are represented as bipartite graphs where existing methods focus on node-based features while largely ignoring edge information. This work addresses that gap by introducing SKEGAT (Sinkhorn-Normalized Edge-enhanced Graph Attention Network), a model that effectively captures both node and edge features in MILP problems. Additionally, the authors propose an "adaptive Regret-Greedy search method" to enhance solution feasibility and address scalability challenges.

The key contributions of the paper include:
1. The introduction of SKEGAT, which incorporates edge information using Sinkhorn normalization for more accurate and stable learning.
2. A novel adaptive Regret-Greedy search algorithm that improves variable assignment strategies, ensuring more accurate and feasible solutions.
3. Experiments on several combinatorial optimization problems (e.g., Combinatorial Auction and Item Placement), showing that SHARP surpasses state-of-the-art (SOTA) solvers and machine learning-based methods, achieving improvements in primal gap and computational efficiency.

**Strengths:**

I am not an expert in this specialized topic. I had to read (from scratch) a lot of background on the development of this subject and within a very short span of time, so my assessment may differ from the experts in this topic. I will also check the comments of the other referees. According to me,
1. The paper introduces the use "edge features" in solving MILP problems, which seem to have been overlooked in prior work. The combination of "SKEGAT" (an Edge-enhanced Graph Attention Network) with "Sinkhorn normalization" is a novel and creative approach that adds value to the MILP solving domain.

2. The paper demonstrates technical rigor, offering both theoretical and empirical contributions. The authors provide justification for the design of SHARP, with clear explanations of the model components and their roles. The results from comprehensive experiments show significant improvements over state-of-the-art solvers, lending credibility to the method's practical utility.

3. The paper is generally well-organized and clearly written. Complex concepts, such as graph attention networks and the Sinkhorn algorithm, are explained in a manner that is accessible to a broad audience. Visual aids like diagrams and flowcharts are effective in clarifying the model’s structure and experimental results.

4. The contribution is in the field of combinatorial optimization and MILP solving, where incorporating edge information represents a step forward. The demonstrated improvements over widely-used solvers like Gurobi and SCIP highlight SHARP’s potential for practical applications.

**Weaknesses:**

The following weaknesses stand out for me (again under the caveat mentioned in the Strengths section)
1. Limited Comparison with Recent MILP Solving Techniques - The paper compares SHARP primarily with traditional solvers like Gurobi and SCIP and a single machine learning-based approach (PaS). However, more recent methods in MILP solving, such as reinforcement learning or hybrid models integrating deep learning with heuristics, have not been considered.

2. Narrow Range of MILP Problems and Model Variants - The experiments are limited to only a few problem types (Combinatorial Auctions, Item Placement). Furthermore, the paper tests SHARP on a limited range of model sizes and problem complexities.

3. Sparse Analysis of Computational Overhead - The paper does not sufficiently address the computational cost of the SHARP framework, particularly the added complexity of SKEGAT and Sinkhorn normalization, which could potentially negate the performance gains in larger-scale settings. A more detailed analysis of SHARP’s computational overhead, including training and inference times compared to other ML-based solvers, would help readers assess its practical viability. Adding a discussion on how SHARP scales with larger datasets and graph sizes would also be beneficial.

4. Lack of Sensitivity Analysis on Hyperparameters - The paper presents fixed hyperparameter settings without exploring their sensitivity or impact on the model’s performance.

**Questions:**

1. What is the state of the art when we incorporate other recent techniques including RL methods?

2. To validate the generalization of SHARP, what happens when we treat other optimization challenges such as Maximum Independent Set, cut selection problem etc ?

---

### Official Review · Reviewer_SEEc · 2024-11-03

**Soundness:** 2
**Presentation:** 2
**Contribution:** 2
**Rating:** 3
**Confidence:** 2

**Summary:**

This manuscript presents SHARP, which draws inspiration from existing frameworks, particularly Light-MILPopt, which can enhance the utilization of edge information and introduce a post-hoc searching technique. SHARP integrates both node and edge features through SKEGAT, enhancing model expressiveness and training stability. The framework also introduces a confidence threshold-based regret greedy search method to improve solution feasibility and accuracy, overcoming scalability limitations.

**Strengths:**

1. The framework employs an adaptive Regret-Greedy search algorithm that leverages marginal probabilities to strategically fix variable values, effectively reducing problem size and addressing the exponential complexity associated with MILP solvers.
2. Through comprehensive experiments, SHARP has shown to outperform both modern MILP solvers like Gurobi and SCIP and some ML-based algorithms in terms of primal gaps, survival rates, and primal integrals.

**Weaknesses:**

1. The authors acknowledge that their method is inspired by [1]; in fact, the proposed SHARP employs a nearly identical GNN framework to Light-MILPopt in [1]. Notably, both methods utilize EGAT and doubly stochastic normalization, with SHARP's sole distinction being the use of Sinkhorn normalization. However, I fail to see any fundamental difference between the two, as both appear capable of incorporating edge information into MILP modeling.

2. Although I was previously unfamiliar with ML-based MILP, I find it unreasonable that the authors draw extensively from the design of Light-MILPopt without (1) including it as a baseline or (2) thoroughly discussing the essential distinctions between the two.

3. The authors’ two core contributions include better usage of edge information, which seems to have already been addressed in [1], and the introduction of a post-hoc processing method, which, as acknowledged by the authors, largely derives from [2]. Thus, what exactly constitutes the authors' core contribution? It seems more akin to a combination of existing solutions.

Minor comments:

1. Line 151 explains the meaning of $W^l$, yet it does not appear in Eq. (3).
2. What does $\boldsymbol{a}^T$ represent in Eq. (4)?
3. The method in Algorithm 1 appears to be an existing algorithm; why is it included in the methodology section rather than the preliminary section?


[1] Light-MILPopt: Solving Large-scale Mixed Integer Linear Programs with Lightweight Optimizer and Small-scale Training Dataset

[2] Confidence Threshold Neural Diving

**Questions:**

see weaknesses

---

### Official Review · Reviewer_Z5sd · 2024-11-04

**Soundness:** 2
**Presentation:** 2
**Contribution:** 2
**Rating:** 3
**Confidence:** 4

**Summary:**

This paper presents incremental advancements to the prediction and search method outlined in [1]. The enhancements include: 1) the introduction of a newly proposed GNN called SKEGAT, which improves upon EGAT by implementing doubly stochastic normalization using the Sinkhorn algorithm; and 2) a modified hyper-parameter for the search component of [1]. Experiments demonstrate certain improvements across three datasets.

**Strengths:**

1. The paper is well-organized and clearly structured.
2. The adaptation of Sinkhorn normalization to EGAT for bipartite graphs is novel.

**Weaknesses:**

1. The evaluation is insufficient. Although the authors report metrics such as primal gap, survival rate, and primal integral, these do not directly illustrate the advantages of SKEGAT over GAT and GCN. The reported improvements may stem from the search module. At a minimum, an additional metric like "accuracy" should be included to demonstrate that SKEGAT yields more accurate predictions.
2. The modification from $\\sum\_{x\\in\\mathcal{X}\_0} x  + \\sum_{x\\in\\mathcal{X}\_1} (1-x)\\leq \\Delta$ in [1] to the proposed $\\sum\_{x\\in\\mathcal{X}\_0} x  + \\sum\_{x\\in\\mathcal{X}\_1} (1-x)\\leq \\lambda|\\mathcal{X}\_0\\cup\\mathcal{X}\_1|$ appears to be a straightforward normalization. The advantages and motivations for this change are not adequately justified.
3. There are quite a few writing issues that need to be addressed:

    - Incorrect statements:
        - The dimensions of matrix multiplications in lines 3-4 of Algorithm 1 do not match.
        - In line 257, should $\textbf{E}^k$ represent "the edge features of the $k$-th layer" instead of "$k$-th layer"?
        - The loss function in line 268 only works for problems with pure binary variables. An additional explanation for the continuous ones is needed.

    - Missing details:
        - What's the dimension of $\hat{\textbf{E}}$ in Section 2.3?
        - What properties does $\textbf{E}$ satisfy in line 144?
        - What's $W^l$ in line 151?
        - What's $a$ in equation (4)?


    - Typos:
        - Sinkhorn($\alpha^l$) -> Sinkhorn($\hat{\alpha}^l$) in equation (5)
        - $x*$ -> $x\^*$ in lines 310 and 312. Besides, please ensure its definition is introduced when first used.

**Questions:**

1. How is the hyper-parameter tuning conducted? This is a crucial process that should be explained in more detail, as different hyper-parameter settings can significantly affect final performance.





[1] Han Q, Yang L, Chen Q, et al. A gnn-guided predict-and-search framework for mixed-integer linear programming[J]. arXiv preprint arXiv:2302.05636, 2023.

---

### Note · Authors · 2024-11-21

I have read and agree with the venue's withdrawal policy on behalf of myself and my co-authors.